# Calculation of Fringe Angle with Enhanced Phase Sensitivity and 3D Reconstruction

**DOI:** 10.3390/s24227234

**Published:** 2024-11-12

**Authors:** Hongyang Wang, Xin He, Zhonghui Wei, Zhuang Lv, Qiwen Zhang, Jun Wang, Jiawei He

**Affiliations:** 1Changchun Institute of Optics, Fine Mechanics and Physics, Chinese Academy of Sciences, Changchun 130033, China; 2University of Chinese Academy of Sciences, Beijing 100049, China

**Keywords:** phase sensitivity, 3D reconstruction, optimal fringe, measurement accuracy

## Abstract

In the field of fringe projection profilometry, phase sensitivity is a critical factor influencing the precision of object measurements. Traditional techniques that employ basic horizontal or vertical fringe projection often do not achieve optimal levels of phase sensitivity. The identification of the fringe angle that exhibits optimal phase sensitivity has been a significant area of research. The present study introduces a novel method for determining the optimal fringe angle, facilitating 3D reconstruction without the need for equipment adjustments. Initially, the optimal fringe is derived through mathematical analysis, and the system’s position within each coordinate system is standardized, leading to the determination of the optimal fringe angle in the world coordinate system. Subsequently, an optimal fringe pattern, akin to that produced by a rotating projector, is generated based on the concept of rotation around a central point, with corresponding adjustments made to the calibration parameters. Finally, the optimal fringe is projected onto the target object for 3D reconstruction, thereby validating the proposed method. The experimental results demonstrate that this approach accurately identifies the optimal fringe angle, significantly enhancing both phase sensitivity and measurement accuracy. The accuracy of the measurement is significantly greater, by an order of magnitude, compared to the traditional method, with the error being approximately 50% of that associated with the currently established improved method.

## 1. Introduction

In the realm of 3D profile measurement, fringe projection profilometry (FPP) utilizing structured light has gained significant popularity due to its uncomplicated setup, rapid measurement speed, and high resolution [1,2,3,4,5,6,7,8,9]. As the techniques of stereovision advance, the spatial configuration among projectors, cameras, and objects being measured has become more adaptable [10,11], leading to a relaxation of traditional optical geometry constraints. Consequently, 3D contour measurement systems based on stereovision have seen a rise in applications [12,13,14,15,16,17]. The phase component plays a critical role in determining measurement accuracy. In recent years, there has been an increasing focus on the phase sensitivity related to the depth of objects. Within FPP, phase sensitivity stands as a pivotal element influencing the resolution and accuracy of 3D surface measurement, which is essential for ensuring measurement precision and serves as a fundamental aspect for optimizing the 3D measurement system [18,19,20,21,22]. If the phase alteration does not correspond sensitively to changes in object depth, even a minor phase discrepancy can result in substantial depth fluctuations, consequently diminishing measurement accuracy. To mitigate errors and achieve heightened measurement precision, maximizing phase sensitivity becomes imperative.

In recent years, there has been a growing emphasis on enhancing phase sensitivity in fringe projection profilometry (FPP) research. Traditional FPP methods typically derive phase information from object surfaces by projecting horizontal or vertical fringe patterns. However, these simplistic fringe patterns may not always be the most effective choice. Li and Zhang [18] introduced a method to produce two sets of fringes near ±π/4 based on the optimal fringe angle, projecting these mutually perpendicular fringes for 3D reconstruction. This method has shown promise in increasing phase sensitivity and enhancing measurement accuracy. Nevertheless, the increase in projected fringe patterns results in longer measurement times compared to traditional methods. Additionally, the full potential of the optimal fringe angle remains underutilized in this method. Similarly, Zhou et al. [22] conducted an analysis on triangulation principles and the optimal fringe angle for stereo vision. They determined the optimal fringe angle by projecting fringe patterns with varying phase changes. Their findings suggested that the optimal angle aligns with the projector-camera baseline and the horizontal direction, known as the system angle. However, discrepancies between the system angle and the optimum fringe angle introduce errors.

The existing methods for determining the optimal fringe angle encounter challenges such as complex measurement procedures and inaccuracies in results. Furthermore, most current techniques that utilize optimal fringe reconstruction necessitate rotating the projector to the corresponding angle, which is cumbersome and may introduce errors. In general, researchers are striving to investigate the optimal fringe angle and phase sensitivity. The accurate determination of the optimal fringe angle to enhance phase sensitivity, as well as the straightforward and convenient application of this optimal fringe in fringe projection profilometry, presents a significant challenge. A novel method has been proposed to tackle the challenges related to low phase sensitivity and inefficiency in calibrating equipment in existing measurement techniques. This method aims to identify the optimal fringe angle and make use of the most suitable fringe for 3D reconstruction without requiring equipment adjustments. In contrast to the conventional FPP methods, this novel method allows for the quick, accurate, and convenient determination of the optimal fringe angle, thereby improving phase sensitivity. By utilizing the best fringes for measurement, regardless of the system’s arbitrary positioning, performance can be optimized, overcoming the difficulties associated with equipment adjustments and low phase sensitivity. As a result, measurement accuracy and efficiency are enhanced.

## 2. Methods

Structured light systems typically comprise projectors, cameras, and electronic computers [23]. The projector emits sinusoidal encoded structured light onto the object’s surface, with the fringe pattern being modulated by the object’s height and captured by the camera. The 3D reconstruction process of the proposed method is illustrated in Figure 1. Initially, the spatial geometry position of the system is established by inversely mapping the positions of the projector and camera in the world coordinate system using the system’s calibration parameters. Subsequently, the optimal fringe angle necessary for 3D reconstruction is computed, and the corresponding optimal fringe pattern is generated. Following this, the projector’s position is fixed, and the calibration parameters are adjusted based on the optimal fringe angle to achieve the virtual rotation of the projector. Ultimately, the fringes at the optimal angle are analyzed to acquire precise 3D information about the object. This method leverages the spatial geometry of the system to determine the optimal fringe angle, thereby enabling high-precision 3D reconstruction of the object’s topography.

### 2.1. FPP System Parameters

Both the camera and the projector adhere to the pinhole model [8] illustrated in Figure 2. In this model, $O_{w}-X_{w}Y_{w}Z_{w}$ represents the world coordinate system, with object X being denoted in the world coordinate system as $\left(X_{w},Y_{w},Z_{w}\right)$. $O_{c}-X_{c}Y_{c}Z_{c}$ signifies the camera coordinate system, where the $Z_{c}$ axis corresponds to the optical axis of the camera. The image coordinate systems, $o_{c1}-xyz_{c}$ and $o_{c2}-stz_{p}$, have their origins at the points of intersection with the optical axis, denoted as $o_{c0}$ and $o_{p0}$, respectively. $o_{c0}-u_{c}v_{c}$ represents the pixel coordinate system, with the origin $o_{c0}$ situated at the upper left corner of the imaging plane. On the other hand, $O_{p}-X_{p}Y_{p}Z_{p}$ stands for the projector coordinate system, with $Z_{p}$ representing the optical axis of the projector. The projector is considered analogous to a camera in this context [24], with similar coordinate systems. The points $x^{p}={\left(x_{p},y_{p}\right)}^{T}$ captured by the projector are transformed from the points $x^{c}={\left(x_{c},y_{c}\right)}^{T}$ in the camera coordinate system using the phase value [25].

$X={\left(X_{w},Y_{w},Z_{w}\right)}^{T}$ represents a point in the world coordinate system which is observed by the camera and corresponds to a pixel $x^{c}={\left(x_{c},y_{c}\right)}^{T}$ in the camera image plane. The imaging process based on the pinhole camera model can be expressed as:(1)$$s\left[\begin{array}{c} u_{c}\\ v_{c}\\ 1 \end{array}\right]=\left[\begin{array}{ccc} f_{x} & \gamma & \begin{array}{cc} u_{0} & 0 \end{array}\\ 0 & f_{y} & \begin{array}{cc} v_{0} & 0 \end{array}\\ 0 & 0 & \begin{array}{cc} 1 & \;0 \end{array} \end{array}\right]\left[\begin{array}{cc} R & T\\ 0 & 1 \end{array}\right]\left[\begin{array}{c} X_{w}\\ Y_{w}\\ \begin{array}{c} Z_{w}\\ 1 \end{array} \end{array}\right]$$
where s is the scale factor, and $(u_{c}$,$\;v_{c}{)}^{T}$ is the pixel coordinates of points on the imaging plane. R is a 3 × 3 rotation matrix, and T is a 3 × 1 translation vector. R and ***T*** are referred to as external parameters responsible for converting the world coordinate point $X={\left(X_{w},Y_{w},Z_{w}\right)}^{T}$ into the camera coordinate system using Formula (2).
(2)$$X_{c}=\left[\begin{array}{cc} R & T\\ 0 & 1 \end{array}\right]X$$

The internal parameter matrix K is responsible for the transformation of points from the camera coordinate system to the imaging plane.
(3)$$K=\left[\begin{array}{ccc} f_{x} & \gamma & u_{0}\\ 0 & f_{y} & v_{0}\\ 0 & 0 & 1 \end{array}\right]$$
where $f_{x}$ and $f_{y}$ represent the focal length (in pixels), γ is the skewness coefficient, and $(u_{0}$,$\;v_{0}{)}^{T}$ denotes the principal point located near the center of the imaging plane. In the realm of camera calibration techniques, Zhang’s calibration method [26] stands out as one of the most commonly employed methods, capable of providing more precise estimates of the camera’s internal and external parameters. Finally, the projection matrix of the camera and projector can be expressed as:(4)$$P^{c}=\left(\begin{array}{cc} \begin{array}{cc} P_{11}^{c} & P_{12}^{c} \end{array} & \begin{array}{cc} P_{13}^{c} & P_{14}^{c} \end{array}\\ \begin{array}{cc} P_{21}^{c} & P_{22}^{c} \end{array} & \begin{array}{cc} P_{23}^{c} & P_{24}^{c} \end{array}\\ \begin{array}{cc} P_{31}^{c} & P_{32}^{c} \end{array} & \begin{array}{cc} P_{33}^{c} & P_{34}^{c} \end{array} \end{array}\right)$$
(5)$$P^{p}=\left(\begin{array}{cc} \begin{array}{cc} P_{11}^{p} & P_{12}^{p} \end{array} & \begin{array}{cc} P_{13}^{p} & P_{14}^{p} \end{array}\\ \begin{array}{cc} P_{21}^{p} & P_{22}^{p} \end{array} & \begin{array}{cc} P_{23}^{p} & P_{24}^{p} \end{array}\\ \begin{array}{cc} P_{31}^{p} & P_{32}^{p} \end{array} & \begin{array}{cc} P_{33}^{p} & P_{34}^{p} \end{array} \end{array}\right)$$
For every point $x^{c}={\left(x_{c},y_{c}\right)}^{T}$ on the camera image, the 3D coordinates can be expressed as:(6)$$\left(\begin{array}{c} x_{w}\\ y_{w}\\ z_{w} \end{array}\right)={\left(\begin{array}{ccc} P_{11}^{c}-P_{31}^{c}x_{c} & P_{12}^{c}-P_{32}^{c}x_{c} & P_{13}^{c}-P_{33}^{c}x_{c}\\ P_{21}^{c}-P_{31}^{c}y_{c} & P_{22}^{c}-P_{32}^{c}y_{c} & P_{23}^{c}-P_{33}^{c}y_{c}\\ P_{11}^{p}-P_{31}^{p}x_{p} & P_{12}^{p}-P_{32}^{p}x_{p} & P_{13}^{p}-P_{33}^{p}x_{p} \end{array}\right)}^{-1}\left(\begin{array}{c} P_{14}^{c}-P_{34}^{c}x_{c}\\ P_{24}^{c}-P_{34}^{c}y_{c}\\ P_{14}^{p}-P_{34}^{p}x_{p} \end{array}\right)$$
where $x_{c}$ and $y_{c}$ represent the horizontal and vertical coordinates of the points within the camera image coordinate system, while $x_{p}$ denotes the horizontal coordinates of the points in the projector image coordinate system.

The acquired parameters elucidate the device’s principle and the transformation relationship of each coordinate system, thereby establishing a groundwork for resolving the optimal angle in future endeavors. It is evident from Formula (6) that the 3D coordinates are significantly influenced by the calibration parameters of both the camera and the projector. Furthermore, the calibration of the projector must be conducted with the assistance of phase information, indicating that phase plays a critical role in determining the accuracy of the measurement results.

### 2.2. Phase Sensitivity

The ratio of phase change to object height change is termed as phase sensitivity [27,28]. As depicted in Formula (7), with a constant phase change, a higher phase sensitivity results in a smaller alteration in the object’s measurement height. This implies that a greater phase sensitivity leads to reduced sensitivity of the measurement error to phase error, consequently enhancing measurement accuracy.
(7)$$S=\frac{d\phi }{dh}=\frac{d\Delta \phi }{dz_{c}}=\left[\begin{array}{cc} \frac{ds}{dz_{c}} & \frac{dt}{dz_{c}} \end{array}\right]\left[\begin{array}{c} 2\pi f\sin\theta \\ 2\pi f\cos\theta \end{array}\right]$$

In Formula (4), the angle between the fringe development direction and the pole line is represented. Phase sensitivity depends on this angle, indicating that in a stationary measurement system, the fringe direction influences sensitivity. We define the angle between the fringe frequency direction and the polar direction as shown in Figure 3.
(8)$$\cos\delta =\frac{\left[\begin{array}{cc} \frac{ds}{dz_{c}} & \frac{dt}{dz_{c}} \end{array}\right]\left[\begin{array}{c} \sin\theta \\ \cos\theta \end{array}\right]}{\sqrt{{\left(\frac{ds}{dz_{c}}\right)}^{2}+{\left(\frac{dt}{dz_{c}}\right)}^{2}}}$$

By combining Formulas (4) and (5), the following expression is obtained.
(9)$$S=2\pi f\sqrt{{\left(\frac{ds}{dz_{c}}\right)}^{2}+{\left(\frac{dt}{dz_{c}}\right)}^{2}}\cos\delta$$

Phase sensitivity, as shown in Equation (6), is proportional to the cosine of the angle. It peaks at 90 degrees, where the fringe is orthogonal to the polar direction, and is zero at 0 or 180 degrees, indicating parallel alignment with the polar line.

Ideally, the system should be linear, but phase errors are unavoidable in practice. As indicated by Formula (4), low phase sensitivity means that even minor phase errors can significantly affect measurement depth and accuracy. Thus, maximizing phase sensitivity is crucial to reducing the impact of phase errors.

### 2.3. The Optimal Fringe Angle Solution

Figure 4 illustrates the spatial configuration of the structured light fringe projection system [29,30], where $O_{c}$ and $O_{p}$ are the optical centers of the camera and projector. The planes $\pi_{p}$ and $\pi_{c}$ represent the fringe pattern plane of the projector and the imaging plane of the camera, respectively. The baseline intersects at points $O_{p}$ and $O_{c}$, which act as poles on these planes. When $N_{1}$ is a point on a body, the points $N_{1}$, $O_{c}$, and $O_{p}$ define the polar plane. The intersections of this plane with $\pi_{p}$ and $\pi_{c}$, denoted as $l_{p}$ and $l_{c}$, are the polar lines on these planes.

Previous studies have shown that the optimal fringe angle depends on the system’s geometric properties. Determining this angle requires understanding the spatial geometry of the camera and projector, achievable by transforming the camera calibration (Formula (2)).
(10)$$X=R_{c}^{-1}X_{c}-T_{c}$$

Points in the camera coordinate system can be transformed into the world coordinate system to determine the camera’s origin $O_{c}={\left(0,0,0\right)}^{T}$.
(11)$$O_{c}^{w}=-T_{c}$$

The camera coordinate system’s origin in the world coordinate system is denoted as $O_{c}^{w}$, and $O_{p}^{w}$ can be determined accordingly.
(12)$$O_{p}^{w}=-T_{p}$$
where $O_{p}^{w}$ represents the position of the origin of the projector coordinate system within the world coordinate system.

Figure 4 illustrates that the line connecting points $O_{c}$ and $O_{p}$, known as the baseline, allows for the derivation of their coordinates in the world coordinate system. This baseline exists in 3D space, with the camera and projector imaging planes perpendicular to the optical axis at distances $f_{c}$ and $f_{p}$ from $O_{c}$ and $O_{p}$, respectively. The following equations apply:(13)$$Z_{c}=R_{c}^{-1}{\left(0,0,1\right)}^{T}-T_{c}$$
(14)$$Z_{p}=R_{p}^{-1}{\left(0,0,1\right)}^{T}-T_{p}$$
where $Z_{c}$ and $Z_{p}$ represent the direction vectors of the optical axis in the camera coordinate system and the projector coordinate system, respectively. The imaging plane $\pi_{p}$ of the projector can be considered as a plane located at a distance $f_{p}$ from $O_{p}$, with $o_{p}$ denoted the central point that is the intersection between the imaging plane of the projector and $Z_{p}$. The position $o_{p}^{w}$ in the world coordinate system and the equation of the imaging plane of the projector can be derived as follows:(15)$$o_{p}^{w}=O_{p}^{w}+f_{p}\times \left(\frac{Z_{p}}{\left\|Z_{p}\right\|}\right)$$
(16)$$Z_{p}\times \left[X-o_{p}^{w}\right]=0$$
where $\left\|Z_{p}\right\|$ represents the Euclidean norm of the vector $Z_{p}$, and $X={\left(x,y,z\right)}^{T}$ denotes a point located on the imaging plane of the projector. The baseline refers to the line connecting $O_{c}$ and $O_{p}$ in space, and the equation is derived as follows:(17)$$X=O_{c}^{w}+t\times \left(O_{p}^{w}-O_{c}^{w}\right),t\epsilon R$$

By combining Equations (13) and (14), the coordinates of the pole can be derived.
(18)$$e=O_{c}^{w}+\frac{\left[{\left(Z_{p}\right)}^{T}\cdot o_{p}^{w}-{\left(Z_{p}\right)}^{T}\cdot O_{c}^{w}\right]}{{\left(Z_{p}\right)}^{T}}\cdot \left(O_{p}^{w}-O_{c}^{w}\right)$$

The reference point on the imaging plane of the projector is defined as the center, denoted as point $o_{p}$. The polar vector $l_{p}$ is then determined from this reference point.
(19)$$l_{p}=e-o_{p}^{w}$$

The angle θ that is required is the angle between the polar line and the s-axis of the coordinate system of the projector plane.
(20)$$\theta =\mathrm{arccot}[\left(l_{p}\cdot \vec{s}\right)/\left(\|l_{p}\times \vec{s}\|\right)]$$
where $\vec{s}$ represents the unit vector that points in the positive direction of the s-axis.

### 2.4. 3D Reconstruction Method Utilizing the Optimal Fringe

The optimal fringe angle θ that is acquired can be applied for 3D reconstruction. The traditional method requires rotating the projector by an angle of θ for projection, causing changes in the spatial locations of the equipment, which introduces a degree of error and increases measurement complexity. This study presents a reconstruction method that removes the necessity of rotating the projector.

A diagonal fringe at angle θ is introduced into the Digital Light Processing (DLP) system and projected, as shown in Figure 5. The inclined angle causes the phase to unwrap according to the fringe’s orientation, making traditional reconstruction methods unsuitable. However, since the imaging effect of a camera projecting θ angle streaks is similar to that of a rotating projector, a correlation can be established to use θ angle fringes for reconstruction without device modifications.

In general, the light intensity function of a sinusoidal grating with horizontal or vertical fringes projected by a projector can be described as:(21)$$I\left(x,y,\delta_{i}\right)=A\left(x,y\right)+B\left(x,y\right)\cos\left[\phi \left(x,y\right)+\delta_{i}\right]$$
where I represents the light intensity function, A denotes the background light intensity, B stands for the modulation amplitude of the fringe, ϕ signifies the phase corresponding to the point of (x,y), and δ represents the moving phase value. The phase shifts of the four grating images using the four-step phase shift method are 0, π/2, π, and 3π/2, respectively. Images can be produced based on the following formula:(22)$$I\left(x,y\right)=255\times \frac{1}{2}\left\{1+\cos\left(2\pi f_{0}x+\delta_{i}\right)\right\}$$
where $f_{0}$ represents the projection fringe frequency, and (x,y) denotes the coordinate value in the pixel coordinate system. To create a diagonal fringe with an angle of θ that replicates the projection effect of rotating the projector, the fringe image should be rotated by an angle of θ around the central point of the projector plane ($s_{0},t_{0}$). The relationship between the coordinates of the point after rotation (x,y) and the coordinates of the point before rotation (X,Y) is as follows:(23)$$X=\left(x-s_{0}\right)\cos\theta -\left(y-t_{0}\right)\sin\theta +x_{0}$$
The fringe associated with the θ angle is derived by integrating Formulas (19) and (20).
(24)$$I\left(x,y\right)=255\times \frac{1}{2}\left\{1+\cos\left(2\pi f_{0}X+\delta_{i}\right)\right\}$$
The light intensities corresponding to each value are as follows:(25)$$I_{0}\left(x,y\right)=A\left(x,y\right)+B\left(x,y\right)\cos\varphi \left(x,y\right)\;$$
(26)$$I_{1}\left(x,y\right)=A\left(x,y\right)-B\left(x,y\right)\sin\varphi \left(x,y\right)\;$$
(27)$$I_{2}\left(x,y\right)=A\left(x,y\right)-B\left(x,y\right)\cos\varphi \left(x,y\right)\;$$
(28)$$I_{3}\left(x,y\right)=A\left(x,y\right)+B\left(x,y\right)\sin\varphi \left(x,y\right)\;$$
The phase at a specific point can be ascertained by concurrently solving Equations (25) to (28):(29)$$\varphi \left(x,y\right)=\arctan\left(\frac{I_{3}-I_{1}}{I_{0}-I_{2}}\right)$$

The phase acquired in this context is the wrapping phase, which falls within the range of (−π,+π). To address the continuous phase distribution, it is imperative to extend the phase.
(30)ϕ(x,y)=φ(x,y)+2πK(x,y)
where ϕ represents the spread phase and K denotes the fringe order determined through a phase spread algorithm [25,31,32,33,34]. The resultant phase values can be utilized for 3D reconstruction.

The projection of a fringe with an angle θ on the projector can be interpreted as rotating the projector by an angle of θ. Consequently, the calibration parameters of the projector will adjust accordingly.
(31)$$\left[\begin{array}{c} X_{p}\\ Y_{p}\\ Z_{p} \end{array}\right]=R_{p}\left[\begin{array}{c} X_{w}\\ Y_{w}\\ Z_{w} \end{array}\right]+T_{p}$$
(32)$$R_{r}=\left[\begin{array}{ccc} \cos\theta & -\sin\theta & 0\\ \sin\theta & \cos\theta & 0\\ 0 & 0 & 1 \end{array}\right]$$

Formula (28) depicts the points in the world coordinate system corresponding to the projector coordinates. By left-multiplying $R_{r}$, the rotation matrix, the resulting expression can be derived:(33)$$R_{r}\left[\begin{array}{c} X_{p}\\ Y_{p}\\ Z_{p} \end{array}\right]=R_{r}\cdot R_{p}\left[\begin{array}{c} X_{w}\\ Y_{w}\\ Z_{w} \end{array}\right]+R_{r}\cdot T_{p}$$
As demonstrated in Equation (30):(34)$$\left[\begin{array}{cc} R_{p}^{\prime } & T_{p}^{\prime } \end{array}\right]=R_{r}\left[\begin{array}{cc} R_{p} & T_{p} \end{array}\right]=\left[\begin{array}{ccc} \cos\theta & -\sin\theta & 0\\ \sin\theta & \cos\theta & 0\\ 0 & 0 & 1 \end{array}\right]\left[\begin{array}{cc} R_{p} & T_{p} \end{array}\right]$$
where $\left[\begin{array}{cc} R_{p}^{\prime } & T_{p}^{\prime } \end{array}\right]$ represents the external parameter following the rotation of the projector by an angle of θ.

By employing the modified projector calibration parameters and the angle fringe reconstruction technique, the option to relocate the device is eliminated, thereby guaranteeing the use of optimal fringe reconstruction.

## 3. Experiments

This study included experimental procedures aimed at validating the findings presented in the preceding sections. The measurement setup primarily comprised a digital projector (DLP4500, 912 × 1140 pixels) and a digital camera (HIKVISION MV-CH050-10UM, 2448 × 2048 pixels, manufactured by HIKVISION in Changchun, China.). The camera was equipped with a lens (MVL-MF1628M-8MP) with a focal length of 16 mm. The distance between the projector and the object under examination was set at approximately 500 mm. For the analysis of fringe patterns, a four-step phase shift was employed to determine the enveloping phase, the three-frequency heterodyne method (f = 76, 70, 65) was utilized to calculate the absolute phase, and the fringe pattern was measured with phase increments of π/2 radians between consecutive frames. A precision plate with a pattern featuring a two-dimensional array of black dots was used as a calibration plate to calibrate the system parameters. The same board used for the object under test was employed for measurements, with the measuring object positioned on the reference plane during the measurement process.

### 3.1. Optimal Angle

After fixing the system position, the PMP system undergoes calibration using a calibration board. The calibration method proposed by Zhang [26] is employed to determine the pose of 20 sets of calibration boards for computation, resulting in the acquisition of necessary calibration parameters. The algorithm presented in this study effectively determines the spatial position of the system, as illustrated in Figure 6.

As depicted in the Figure 6, $O_{w}$ represents the origin point of the world coordinate system situated at the lower right corner of the calibration board. The surface of the plate is defined as the $X_{w}-Y_{w}$ plane, and the world coordinate system is defined for $Z_{w}$ when the vertical surface of the plate is facing upwards. The polar line $l_{p}$ is determined by extending the intersection point between the image plane and the baseline of the projector to the pole $e_{p}$. Through this method, the optimal fringe angle of the current system is computed to be 62.3751°.

To validate the accuracy of the calculated angle, the angle measurement method proposed by Wang and Zhang [21] was compared. Subsequently, the angle with the highest phase sensitivity was determined through the fitting of various angle projections, which served as the standard angle. The process and outcomes of determining the optimal fringe angle by measuring the standard block are illustrated in Figure 7. Initially, horizontal and vertical fringes were utilized to illuminate the flat plate. Subsequently, the vertical and horizontal fringe patterns were projected onto the standard block (with a height of 50 mm). The phase difference diagram was obtained by subtracting the absolute phase between the standard block and the reference plane captured by the camera. The subtraction of the upper and lower surfaces of the phase difference diagram yielded the vertical fringe phase difference $\Delta \phi_{v}$ and horizontal fringe phase difference $\cdot \phi_{h}$. By dividing these two phases and calculating the arctangent value, the optimal fringe angle was determined to be 60.4377°.

The horizontal and vertical fringe test results indicate that the standard block height for the horizontal fringe test is 43.54 mm, while for the vertical fringe test, it is 51.18 mm. While these results may not serve as a precise assessment of measurement accuracy, they highlight significant variations in the test results across different angle fringes. To minimize measurement errors and enhance accuracy, it is advisable to utilize the optimal angle fringe.

In order to determine the optimal fringe angle of the current system, the fringe was projected at various angles for 3D measurements. The phase difference of the same group of plates and standard blocks under fringes at different angles was recorded, as presented in Table 1. The phase difference reached its maximum at approximately 60°, indicating the highest phase sensitivity at this point, making it the optimal fringe angle. Conversely, at around 150°, the phase difference was minimal, nearly 0, signifying the lowest phase sensitivity and, hence, the least favorable fringe angle. Based on the data, the least favorable fringe angle fell between 150°–160°, while the optimal fringe angle lay between 60°–70°. Curve fitting analysis was conducted on the data to identify the angle with the largest phase difference. To determine the optimal fringe angle more precisely, measurements were taken at 1° intervals within the range of 60°–70°. Figure 8 illustrates that the fringe angle is approximately 62.82° when the phase difference peaks. Table 2 compares the 60.4377° obtained through the standard block method with the 62.3749° measured using the method described in this paper, showing a significant deviation from the angle derived from standard block measurement compared to 62.82°. The method proposed in this paper yields a result closer to the true optimal angle, offering a more efficient and accurate measurement method.

### 3.2. Validation of 3D Reconstruction Accuracy

To investigate the impact of phase error and phase sensitivity on measurement accuracy, various techniques were employed to measure a standardized plate. A line was selected at the center of the measurement plane to analyze the fluctuation of the depth curve, as illustrated in Figure 9a–e. The analysis revealed that the primary factors influencing the depth curve were phase error and phase sensitivity. Notably, the horizontal fringe curve exhibited the highest fluctuation, indicating a low phase sensitivity. Comparatively, the proposed method demonstrated the least curve fluctuation among all methods, suggesting that it can achieve the highest phase sensitivity.

To accurately evaluate the measurement precision of the optimal fringe, horizontal fringe, vertical fringe, Li and Zhang’s method [18], and Wang and Zhang’s method [21], a series of measurements was utilized. Initially, a standard flat plate was measured, followed by the placement of measuring blocks with heights of 5 mm, 10 mm, and 15 mm (with a height error of less than 0.0001 mm). The height of each measuring block was determined by comparing the measurement results of the blocks with the result obtained from the initial flat plate measurement. In the proposed methods, the height of the measuring block was calculated as the average distance between points on a specific surface area in two sets of measurement results.

The measurement results are presented in Table 3. In the current system, there is a significant deviation of the transverse fringe angle from the optimal angle, resulting in an average measurement error of 9.16%. The error in the vertical fringe is slightly better than that in the horizontal fringe, with an average measurement error of 3.803%. The optimization methods proposed by Li and Zhang, as well as Wang and Zhang, have notably enhanced the measurement accuracy, leading to average measurement errors of 0.996% and 0.878%, respectively. The average measurement error using the proposed method is 0.464%. It is evident that utilizing the optimal fringe can effectively enhance measurement accuracy. The algorithm introduced in this study demonstrates improved accuracy in determining the optimal angle without the need for equipment adjustments for reconstruction, thereby further enhancing accuracy.

After determining the optimal fringe angle, the 3D reconstruction process could be initiated utilizing this optimal fringe angle. To validate the effectiveness of the reconstruction, the 3D reconstruction method proposed in the preceding section was employed. The reconstruction was conducted using horizontal fringes, vertical fringes, the Li and Zhang method, the Wang and Zhang method, and the proposed method. It is important to note that the entire hardware setup remained consistent across all experiments. The object remained in the same position, and the fringe period was constant, ensuring that the 12-step phase shift reconstruction result, recalibrated after adjustment to the optimal fringe, was the actual result. Figure 10a–f display the 3D reconstruction outcomes of the various methods. The reconstruction results indicate that the object’s shape was restored when horizontal or vertical fringes were utilized for reconstruction. However, significant errors were observed when compared to the actual result, with the reconstructed outcomes lacking clarity and detail. The optimized reconstruction method can capture more phase information, enhance phase sensitivity, and produce a more accurate reconstructed model with clearer details, resulting in the best reconstruction effect. The reconstruction effects depicted in Figure 10c–e closely resemble the actual result, offering a visually appealing representation. To further assess the accuracy of each method, a specific section of the object under examination was selected for comparison. As illustrated in Figure 10g, the reconstruction outcomes of the three optimization algorithms closely approximated the actual result compared to the results obtained using horizontal and vertical fringes. The section curve of the proposed method aligned most closely with the actual result, demonstrating the superior reconstruction effect on the object’s contour. It is evident that utilizing the optimal fringe determined by the proposed method maximized phase sensitivity, leading to the highest precision and effectiveness in 3D reconstruction.

## 4. Conclusions

This study introduces a novel method designed to enhance phase sensitivity in fringe projection profilometry for enhanced measurement precision. The conventional practice of employing horizontal or vertical fringe projection may not yield optimal results. Therefore, this paper proposes a methodology to determine the optimal fringe angle based on system calibration parameters and subsequently generate the most suitable fringe patterns. The experimental findings indicate that the accuracy of the measurement is significantly greater, by an order of magnitude, when compared to the traditional method. Furthermore, the associated error is approximately 50% of that linked to the currently established improved methods.

The method offers the advantage of calculating the optimal angle based on system parameters, eliminating the need for equipment adjustments. This reduces the complexity and time required for the experiment. The efficacy and precision of the method are confirmed through the utilization of the optimal angle fringe for 3D reconstruction and subsequent measurement verification. This holds significant value for applications demanding precise measurements in industrial and scientific domains.

Future research endeavors may delve into investigating the suitability of this method in various contexts and consider integrating additional technical methods to enhance measurement precision. Furthermore, there is potential for optimizing and examining the method and accuracy of system parameter calibration to enhance the overall performance of the measurement system. Future research efforts should focus on overcoming these limitations and developing more robust and efficient techniques for improving phase sensitivity in FPP systems.

## Figures and Tables

**Figure 1 sensors-24-07234-f001:**
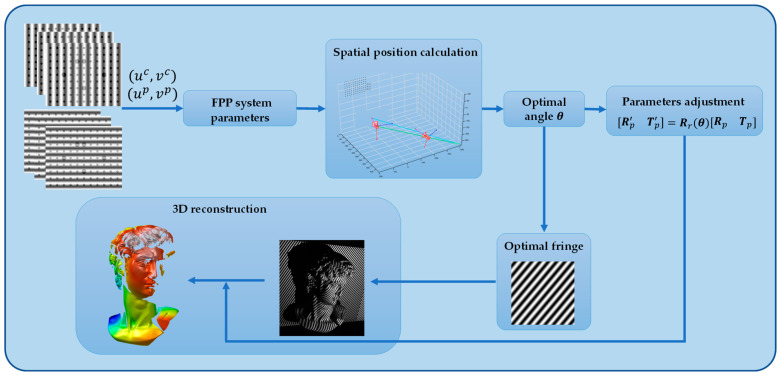
The process of 3D reconstruction in the proposed method.

**Figure 2 sensors-24-07234-f002:**
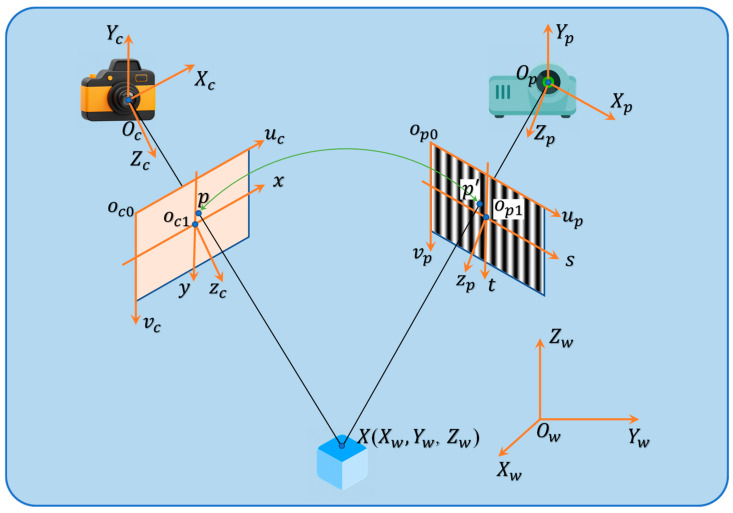
Reconstruction principles and parameters of structured light.

**Figure 3 sensors-24-07234-f003:**
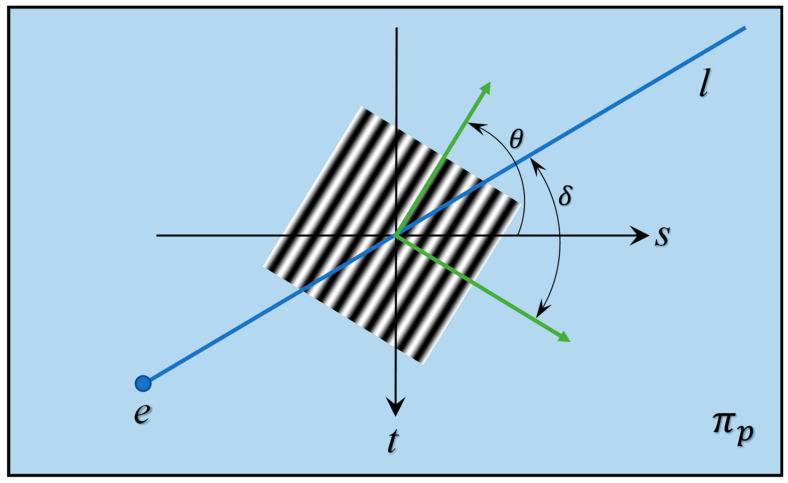
The phase sensitivity is contingent upon the angle formed between the fringe direction and the pole line.

**Figure 4 sensors-24-07234-f004:**
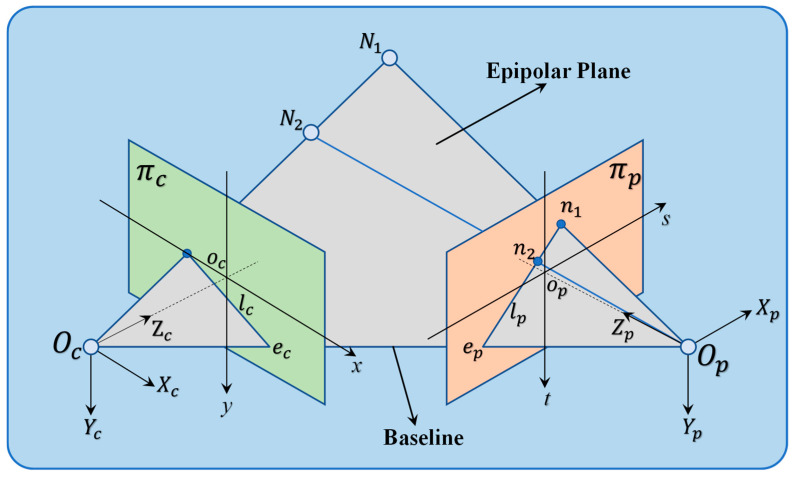
Schematic diagram illustrating the polar principle.

**Figure 5 sensors-24-07234-f005:**
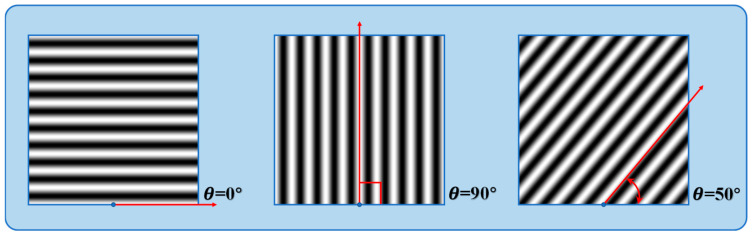
Diagram depicting the fringe angle.

**Figure 6 sensors-24-07234-f006:**
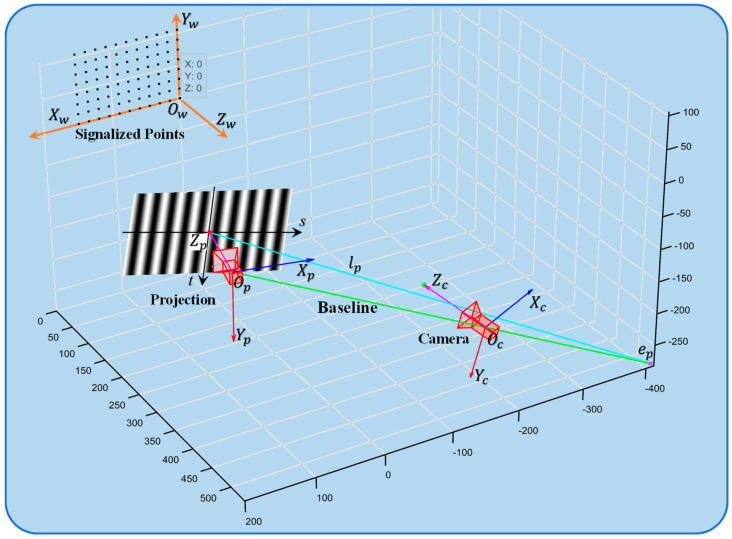
The spatial relationships of various components within the system in the world coordinate system.

**Figure 7 sensors-24-07234-f007:**
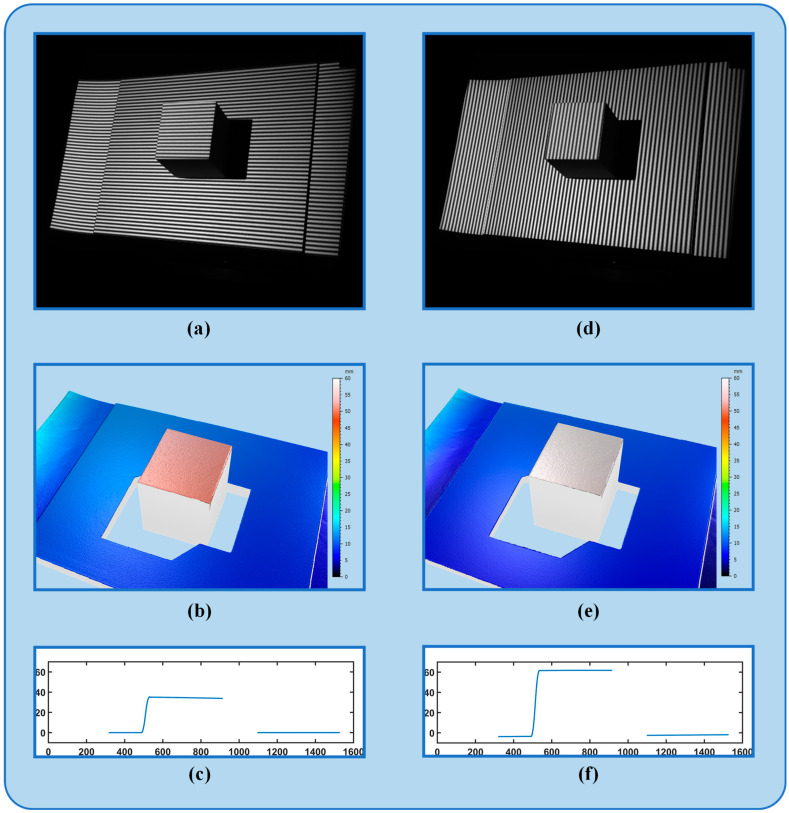
(**a**) Measurement of horizontal fringe. (**b**) Results of horizontal fringe reconstruction. (**c**) Unwrapping phase of the middle line in horizontal fringe measurements. (**d**) Measurement of vertical fringe. (**e**) Results of vertical fringe reconstruction. (**f**) Unwrapping phase of the middle line in vertical fringe measurements.

**Figure 8 sensors-24-07234-f008:**
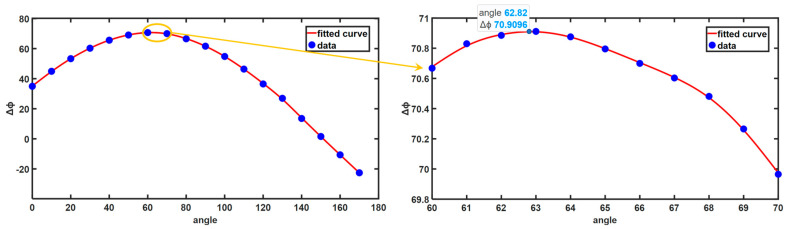
The curve illustrates the fitting of phase differences at different angles.

**Figure 9 sensors-24-07234-f009:**
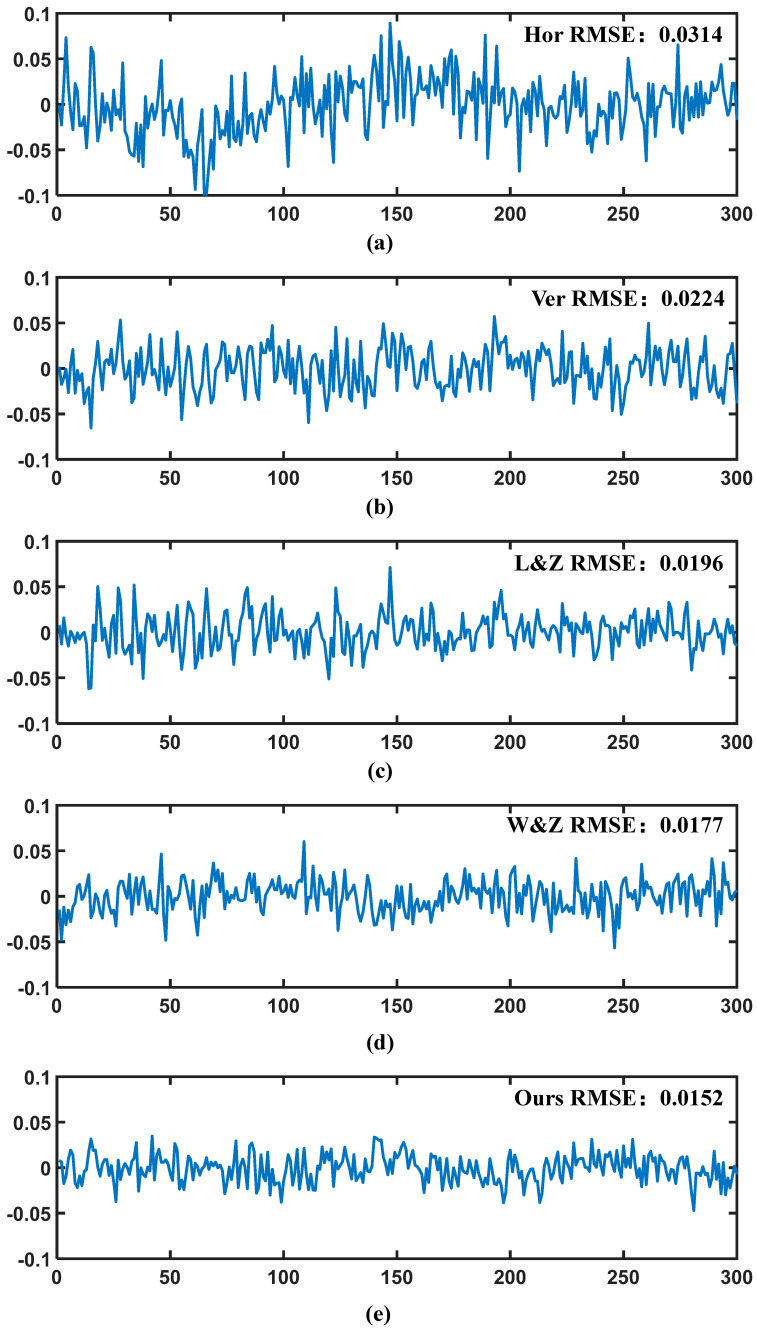
(**a**) Fluctuation in measurement error of horizontal fringe. (**b**) Fluctuation in measurement error of vertical fringe. (**c**) Fluctuation in measurement error of Li and Zhang’s method. (**d**) Fluctuation in measurement error of Wang and Zhang’s method. (**e**) Fluctuation in measurement error of the proposed method.

**Figure 10 sensors-24-07234-f010:**
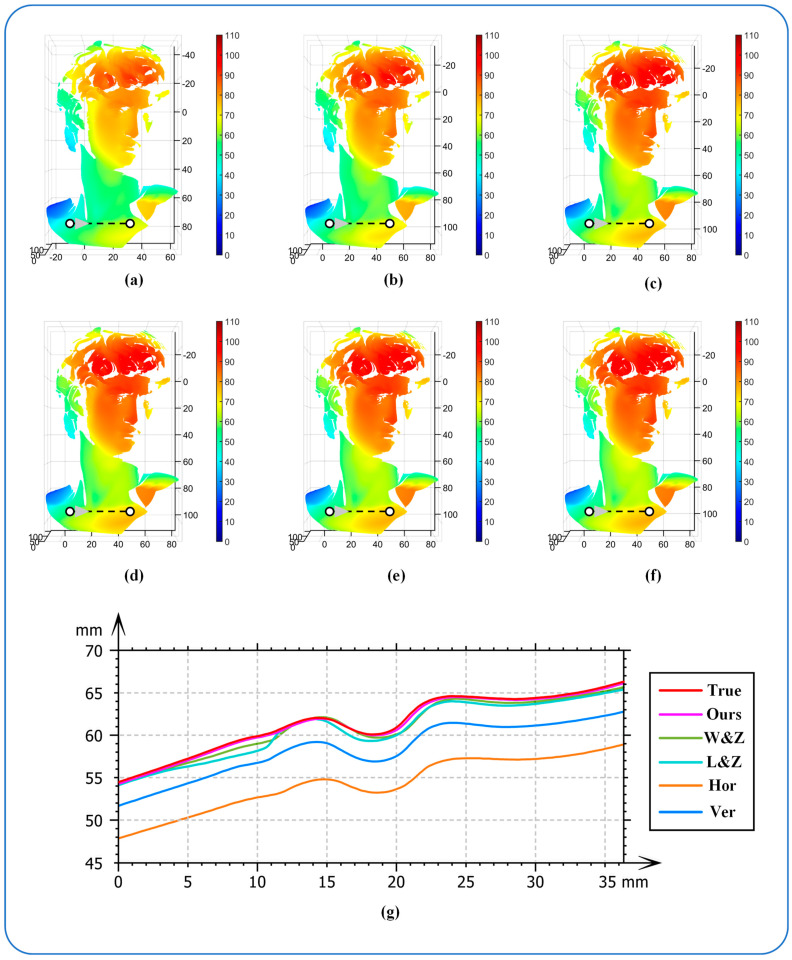
(**a**) Horizontal fringe reconstruction. (**b**) Vertical fringe reconstruction. (**c**) Li and Zhang reconstruction. (**d**) Wang and Zhang reconstruction. (**e**) Our reconstruction. (**f**) The actual reconstruction. (**g**) A sectional comparison map.

**Table 1 sensors-24-07234-t001:** The depiction illustrates the phase difference at various angles. The data highlighted in bold correspond to the fringe angle, while the phase difference is presented below.

**0** **°**	**10** **°**	**20** **°**	**30** **°**	**40** **°**	**50** **°**	**60** **°**	**70** **°**	**80** **°**
34.9477	44.9141	53.3111	60.2608	65.5912	69.0441	70.6678	69.965	66.5330
**90** **°**	**100** **°**	**110** **°**	**120** **°**	**130** **°**	**140** **°**	**150** **°**	**160** **°**	**170** **°**
61.6134	54.7897	46.3797	36.4866	26.9502	13.5368	1.5619	−10.6555	−22.5894

**Table 2 sensors-24-07234-t002:** Angles measured utilizing each method.

Block Method	Curve Fitting Method	Ours
60.4377°	62.82°	62.3751°

**Table 3 sensors-24-07234-t003:** Results of each method test.

Method	Horizontal	Vertical	Li and Zhang	Wang and Zhang	Ours	The Truth
Value	4.2204	5.1827	4.9235	5.0644	4.9697	5.000
	9.3126	10.4204	9.9164	10.0792	10.0452	10.000
	14.2485	15.5326	14.9067	15.0831	15.0502	15.000
Mean error	9.159%	3.803%	0.996%	0.878%	0.464%	—

## Data Availability

The datasets generated and/or analyzed during the current study are available from the corresponding author upon reasonable request.

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
