# Peer review of "Calculation of Fringe Angle with Enhanced Phase Sensitivity and 3D Reconstruction"

_sensors, 2024, doi:10.3390/s24227234_

Round 1
Reviewer 1 Report
Comments and Suggestions for Authors
The paper is connected with calculation of fringe angle with enhanced phase sensivity. This is quite interesting from the point of studies of phase sensivity, finding optimal fringe angles and another applied problems.
First of all, the authors give a short introduction to this field, describing history of the problem and some perspectives. After that they pass to description of the methods and give some geometrical laws that show the connection between different coordinate systems. Then they speak about phase sensivity and find the optimal fringe angle solution. After that they make a 3D reconstruction of the optimal fringe. The next part of the paper is connected with experimental tests of the setup and comparison with theory. In the end, the authors pass to the conclusion.
There are some serious problems of the research:
- the description of the angles and coordinate systems seems to be very simple. It is based on secondary school geometry and should be shortened;
- also, it would be useful to give some links to mathematical books in this field;
- fig.8 shows fitting of the experimental points. It would be useful to give a theoretical curve
I think, that such problems should be solved to publish the paper
Author Response
Response to Reviewer 1 Comments
Dear Reviewer,
We would like to thank you for your efforts in reviewing our manuscript titled "Calculation of Fringe Angle with Enhanced Phase Sensitivity and 3D Reconstruction", and providing many helpful comments and suggestions, which will all prove invaluable in the revision and improvement of our paper, as well as in guiding our research in the future.
Point 1: The description of the angles and coordinate systems seems to be very simple. It is based on secondary school geometry and should be shortened.
Response 1: I fully concur with your suggestion.This paper introduces a straightforward and efficient method for determining the optimal fringe angle, which does not rely on complex mathematical theories but effectively addresses the problem. In response to your request, I have revised sections 2.2, 2.3, and 2.4 of the theoretical framework to provide a more concise and clear description of the angle and coordinate systems. I hope these modifications enhance your understanding of the material. The changes have been highlighted in red.
Point 2: It would be useful to give some links to mathematical books in this field.
Response 2: Your suggestion is highly reasonable. I have incorporated the following two articles on system architecture into the references, with the intention of enhancing readers' understanding. Additionally, if you have any superior references that you could recommend, I would be pleased to include.
- Vargas, R.; Romero, L.A.; Zhang, S.; Marrugo, A.G. Calibration method based on virtual phase-to-coordinate mapping with linear correction function for structured light system. Optics and Lasers in Engineering 2024, 183, 108496.
- Xu, X.; Fei, Z.; Yang, J.; Tan, Z.; Luo, M. Line structured light calibration method and centerline extraction: A review. Results in Physics 2020, 19, 103637.e them.
Point 3: Fig.8 shows fitting of the experimental points. It would be useful to give a theoretical curve.
Response 3: Your suggestion is quite commendable; however, there are certain challenges that may arise in its practical implementation.According to the definition of phase sensitivity , the phase sensitivity remains undetermined. When measuring the same object , which is a constant value, the magnitude of phase sensitivity can be inferred from the size of ; specifically, a larger corresponds to a greater phase sensitivity . Figure 8 illustrates the measurement of the phase difference at intervals of 10 degrees for the fringe angle, followed by more precise measurements at 1-degree intervals near the optimal angle. Curve fitting of the data points yields the angle associated with the maximum , identified as the optimal angle. It is not difficult to find that this method is very troublesome to solve the optimal angle, but it has a high accuracy.The measurement results presented in Figure 8 serve to validate the accuracy of the proposed method. Currently, there is a lack of relevant data to directly derive the theoretical curve of in relation to the angle ; therefore, this study employs an experimental point fitting approach. Should theoretical curves be established in future research, it is anticipated that the method for determining the optimal angle could be further streamlined and enhanced in efficiency. This may represent a potential future direction for research within this field.
Thank you again for your valuable comments and suggestions.
Yours sincerely,
Hongyang Wang

Reviewer 2 Report
Comments and Suggestions for Authors
This paper proposes a method to enhance phase sensitivity in fringe projection profilometry for improved measurement precision. The topic is interesting, and the writing is relatively well. I recommend publishing the work in a journal with some revisions:
(1) Line 20-22, In the abstract section, the results should be described using quantitative measures to illustrate the enhancement in method accuracy. Similarly, the results should be modified in the conclusion section to reflect this quantitative approach.
(2) Line 57-66, The scientific problem should be highlighted in the introduction section. The introduction section requires careful attention.
Specific comments:
(1) Line 64, The statement "Future research..." appears to be unreasonable and should be relocated to the discussion section or rewritten.
(2) Line 79, Figure 1 does not provide sufficient informative content; I suggest redrawing the figure (reference: https://doi.org/10.3390/rs13204165).
(3) Line 130, all the terms in the equation, such as Xc, Yc, and Xp, should be clearly defined.
(4) line 376, the references for "Li and Zhang's method" and "Wang and Zhang's method" should be appropriately included.
Author Response
Response to Reviewer 2 Comments
Dear Reviewer,
We would like to thank you for your efforts in reviewing our manuscript titled "Calculation of Fringe Angle with Enhanced Phase Sensitivity and 3D Reconstruction", and providing many helpful comments and suggestions, which will all prove invaluable in the revision and improvement of our paper, as well as in guiding our research in the future.
Point 1: Line 20-22, In the abstract section, the results should be described using quantitative measures to illustrate the enhancement in method accuracy. Similarly, the results should be modified in the conclusion section to reflect this quantitative approach.
Response 1: I have revised the manuscript in accordance with your comments. The accuracy is quantitatively detailed in lines 22-24 and 426-429.
Point 2: Line 57-66, The scientific problem should be highlighted in the introduction section. The introduction section requires careful attention.
Response 2: I have revised the paragraph by positioning the scientific problem at the beginning of the final paragraph of the introduction. This adjustment serves to emphasize the significance of the scientific problem and to capture the reader's attention more effectively.
Point 3: Line 64, The statement "Future research..." appears to be unreasonable and should be relocated to the discussion section or rewritten.
Response 3: I have relocated this section to lines 440-442 in accordance with your comments.
Point 4: Line 79, Figure 1 does not provide sufficient informative content; I suggest redrawing the figure (reference: https://doi.org/10.3390/rs13204165).
Response4: I have redrawn Figure 1 and incorporated additional information to enhance the reader's comprehension.
Point 5: Line 130, all the terms in the equation, such as Xc, Yc, and Xp, should be clearly defined.
Response5: The definitions for Xc, Yc, and Xp have been incorporated on lines 132-134.
Point 6: line 376, the references for "Li and Zhang's method" and "Wang and Zhang's method" should be appropriately included.
Response6: I have incorporated the reference identifier on line 367 in accordance with your comments.
Thank you again for your valuable comments and suggestions.
Yours sincerely,
Hongyang Wang

Round 2
Reviewer 1 Report
Comments and Suggestions for Authors
Thank you for taking into account my remarks. One of the problems connected with simple description of school mathematics still remains. However, this part has been made smaller
Reviewer 2 Report
Comments and Suggestions for Authors
All issues I proposed have been well addressed. I recommend publishing the manuscript in the journal in its current form.